# Tumor Promoting Effects of Sulforaphane on Diethylnitrosamine-Induced Murine Hepatocarcinogenesis

**DOI:** 10.3390/ijms23105397

**Published:** 2022-05-12

**Authors:** Jie Zheng, Do-Hee Kim, Xizhu Fang, Seong Hoon Kim, Soma Saeidi, Su-Jung Kim, Young-Joon Surh

**Affiliations:** 1College of Pharmacy, Seoul National University, Seoul 08826, Korea; whitecheek@snu.ac.kr (J.Z.); heesoo123@snu.ac.kr (X.F.); daduhoon@snu.ac.kr (S.H.K.); 2Department of Chemistry, College of Convergence and Integrated Science, Kyonggi University, Suwon 16627, Korea; dohee@kyonggi.ac.kr; 3Department of Molecular Medicine and Biopharmaceutical Science, Graduate School of Convergence Science and Technology, Seoul National University, Seoul 08826, Korea; saeidi@snu.ac.kr (S.S.); nynna79@snu.ac.kr (S.-J.K.); 4Cancer Research Institute, Seoul National University, Seoul 03080, Korea

**Keywords:** NRF2, sulforaphane, hepatocellular carcinoma, diethylnitrosamine, chemoprevention

## Abstract

Nuclear factor erythroid 2-related factor 2 (NRF2) is a key transcription factor involved in protection against initiation of carcinogenesis in normal cells. Notably, recent studies have demonstrated that aberrant activation of NRF2 accelerates the proliferation and progression of cancer cells. The differential effects of NRF2 on multi-stage carcinogenesis have raised a concern about the validity of NRF2 activators for chemoprevention. This prompted us to assess the effects of sulforaphane (SFN), a prototypic NRF2 activating chemopreventive phytochemical, on experimentally induced carcinogenesis. In the present study, SFN was daily injected intraperitoneally (25 mg/kg) for 3 months to male C57BL/6 mice at 6 months after single intraperitoneal administration of a hepatocarcinogen, diethylnitrosamine (DEN). The liver to body weight ratio, tumor growth, and the number and the size of hepatomas measured at 9 months after DEN administration were significantly higher in SFN-treated mice than those in vehicle-treated mice. Moreover, the expression of NRF2, its target protein NAD(P)H:quinone oxidoreductase 1, and the cell proliferation marker, proliferating cell nuclear antigen was further elevated in DEN plus SFN-treated mice. These results suggest that once hepatocarcinogenesis is initiated, SFN may stimulate tumor progression.

## 1. Introduction

Hepatocellular carcinoma (HCC), the predominant type of primary liver cancer, is the fourth most common cause of cancer-related mortality globally and occurs in patients with chronic liver diseases, alcohol abuse, and environmental exposure to hepatocarcinogens [1]. Diverse putative driver mutations and oncogenic pathways involved in HCC development have been identified with next-generation sequencing analyses [2,3,4,5,6,7,8,9]. Diethylnitrosamine (DEN) is the most widely used chemical carcinogen to study the biology of HCC. A single intraperitoneal (i.p.) administration of DEN (25 mg/kg) is sufficient to induce liver tumor formation after 8–10 months in 100% of male mice but only 10–30% of female mice [10].

NRF2 is a key transcription factor that activates the cellular stress response to protect cells from carcinogenic and other assaults [11]. In this context, NRF2 has been typically considered a tumor suppressor because of its contribution to host anticancer defense mechanisms. However, multiple lines of increasing evidence suggest that NRF2 can also promote the survival and growth of cancer cells [12]. NRF2 directly or indirectly modulates cellular signaling involved in inflammation, metabolic reprogramming, xenobiotic efflux, and cell proliferation, as well as cellular resistance to oxidative stress [13,14]. Due to such Janus-face nature of NRF2, its role in multi-stage carcinogenesis has generated controversy [14,15,16].

In homeostatic conditions, NRF2 interacts with Keap1, a substrate adaptor protein for the Cul3-dependent E3 ubiquitin ligase complex, which represses NRF2 by promoting ubiquitination and subsequent proteasomal degradation [11]. When cells are challenged with oxidative stress and other stimuli, NRF2 dissociates from Keap1 and translocates to the nucleus where it stimulates the transactivation of antioxidant and other cytoprotective genes [11,14]. Dysregulation of Keap1-NRF2 signaling, such as loss-of-function of Keap1 and gain-of-function of NRF2 through somatic mutation, accounts for constitutive activation of NRF2. Mutations of NRF2 have been frequently found in 3–7% of HCC patients [2,3,4]. Moreover, somatic mutations of NRF2 were noticed in the early stage of rat hepatocarcinogenesis [17]. Notably, the development of preneoplastic lesions was fully prevented by the genetic inactivation of NRF2 in rat and mouse models of hepatocarcinogenesis [18,19]. In addition, NRF2 exerts an oncogenic function via de-glycation by fructosamine-3 kinase in HCC [20]. These findings indicate that NRF2 may act as a driver of HCC development.

The differential effects of NRF2 on multi-stage carcinogenesis have raised a concern about the validity of NRF2 inducers/activators for chemoprevention. Sulforaphane (SFN) has been known to be a powerful inducer/activator of NRF2. The majority of health-beneficial effects of broccoli consumption have been considered to be mediated by SFN. However, SFN is present in broccoli and other cruciferous vegetables as its precursor, glucoraphanin, and is released through the action of myrosinase, a β-thioglucosidase present in either the plant tissue or the gut microbiome after intensive chewing or during digestion, respectively [21]. Despite the well-known chemopreventive potential of SFN, pro-oncogenic effects of this prototypic phytochemical with pronounced NRF2 inducing activity have been reported in lung cancer and colorectal cancer [22,23]. However, the effects of SFN given during the post-initiation period of tumorigenesis remain poorly resolved. In the previous study, we demonstrated the role of NRF2 in chemically-induced liver cancer development in mice [19]. This prompted us to assess the effect of SFN on DEN-induced murine hepatocarcinogenesis.

## 2. Results

### 2.1. The Expression of NRF2 Was Markedly Elevated in DEN-Induced Hepatocarcinogenesis

Two-week-old male mice were given a single intraperitoneal dose of 25 mg/kg DEN prepared in phosphate-buffered saline (PBS; pH 7.4), following the standard protocol. Livers were isolated at the 12th month after DEN administration. The liver to body weight ratio and the tumor volume were significantly higher in DEN-treated mice than in vehicle-treated mice (Figure 1A). Hematoxylin and eosin (H&E) staining clearly distinguished the tumor and adjacent normal tissues (Figure 1B). The expression of NRF2 was markedly elevated in the liver of DEN-treated mice (Figure 1C).

### 2.2. B-Raf ^V637E^ Mutation Was Detected in DEN-Induced Liver Tumors

Somatic mutation of the *B-Raf* proto-oncogene has been observed in patients with advanced HCC [24,25] and is associated with poor prognosis and increased risk of recurrence [26]. This prompted us to measure the *B-Raf* mutation profile in DEN-induced murine hepatocarcinogenesis. Primers for sequencing are listed in Appendix A. Alterations of amplicons from the PCR process using designed primers were detected by capillary electropherogram (Appendix A). Of the 7 hepatoma samples from DEN-treated mice, 4 cases of *B-Raf* ^V637E^ mutation (*n* = 7) were detected (Figure 1D). All the specimens provided (*n* = 14) meet the requirement of quality value (QV > 35), and detailed information of all specimens is listed in Appendix A (NCBI reference sequence: NP_647455.3).

### 2.3. SFN Treatment Promoted Tumor Growth in a DEN-Induced Murine Hepatocarcinogenesis Model

SFN was daily injected intraperitoneally (25 mg/kg) for 3 months to male C57BL/6 mice at the 6th month after the DEN administration (Figure 2A). Livers were isolated at the 9th month after DEN administration (Figure 2B). SFN administration further increased the liver to body weight ratio, the tumor volume, and the tumor number compared to those in DEN-treated mice (Figure 2C). Three out of four cases of *B-Raf* ^V637E^ mutation were detected by capillary electrophoresis in SFN plus DEN-induced liver tumors (Appendix A).

### 2.4. The Expression Levels of NRF2, NAD(P)H: Quinone Oxidoreductase 1 (NQO1), and Proliferating Cell Nuclear Antigen (PCNA) Were Further Elevated by SFN Administration in DEN-Treated Mice

The Western blot analysis revealed markedly elevated expression of NRF2, NQO1, and PCNA in DEN-treated mice. SFN treatment further enhanced the expression levels of NRF2, NQO1, and PCNA in DEN-treated mice (Figure 3A). However, there were no significant differences in the expression levels of Keap1 between DEN-treated and vehicle-treated mice. Corresponding quantifications of protein levels were determined with Image J software (Figure 3B). These effects of SFN were verified by immunohistochemical (IHC) analysis (Figure 4).

### 2.5. SFN Treatment Enhanced the Nuclear Translocation of NRF2 in DEN-Treated Mice

NRF2 activation requires its dissociation from the inhibitory protein, Keap1, and subsequent localization to the nucleus. Cytosol and nuclear extracts of livers were subjected to Western blot analysis. Nuclear protein levels of NRF2 were markedly elevated in the livers of DEN-treated mice compared with those in PBS-treated control mice. The migration of NRF2 to the nucleus was further enhanced by SFN administration in DEN-treated mice. Administration of SFN alone also induced nuclear accumulation of NRF2. However, there was no substantial difference in cytosolic protein levels of NRF2 between DEN- and vehicle-treated mice. (Figure 5).

### 2.6. SFN Treatment Markedly Elevated the Expression Levels of NQO1 in DEN-Treated Mice

Immunofluorescence staining clearly revealed that the expression levels of NQO1 were markedly elevated in DEN-treated mice. SFN treatment further escalated the expression levels of NQO1 in DEN-treated mice (Figure 6).

## 3. Discussion

Cancer chemoprevention refers to the administration of nontoxic natural, synthetic or biological agents to inhibit, retard, or even reverse the multi-step tumorigenesis [27]. SFN, a promising chemopreventive phytochemical, has been reported to interfere with experimentally induced carcinogenesis. Numerous studies have reported that SFN exerts chemopreventive effects in various tumor models through induction/activation of NRF2 signaling [28,29,30,31,32,33]. One of the well-defined mechanisms underlying SFN-induced activation of NRF2 and consequent transactivation of its target genes involves covalent modification of specific sensor cysteine residues of Keap1 [34,35,36]. The interaction between SFN and thiol groups of Keap1 yields thionoacyl adducts, which enables NRF2 to escape from Keap1-mediated ubiquitination and subsequent proteasomal degradation, leading to nuclear localization of NRF2 [27,37,38,39].

NRF2 is well-known as the key regulator of cellular redox balance, and NRF2-mediated expression of antioxidant enzymes contributes to the maintenance of redox homeostasis. Moderate amounts of reactive oxygen species (ROS) can promote pro-tumorigenic signaling, facilitating cancer cell progression [40]. However, aberrant ROS accumulation triggers oxidative stress-induced cancer cell death [40,41]. The collapse of NRF2-mediated antioxidant signaling gives rise to elevated ROS levels and oxidative DNA damage during tumorigenesis [42]. In this context, NRF2 protects tumor tissues from oxidative damage, thereby preventing cancer cell death. Augmentation of NRF2 signaling by SFN treatment is hence anticipated to rescue cancer cells from oxidative DNA damage, which in turn promotes tumor growth and survival in DEN-induced hepatocarcinogenesis.

The DNA damage sensor γ-H2AX was found to be overexpressed in preneoplastic lesions of HCC and considered a useful biomarker for predicting the risk of HCC [43]. ROS at low levels may cause mutations and genomic instability through DNA double-strand breaks (DSBs) [44]. DSBs can be lethal to a cell, and cause apoptosis if not repaired. It was reported that SFN induced DSBs in HeLa cells, subsequently resulting in cancer cell death [45]. However, other studies have demonstrated that SFN seems to protect against DNA damage [46,47]. Both normal tissues and tumor tissues may benefit from such protective effect of SFN, resulting in tumor preventive and tumor promotive effects, respectively. In line with this notion, we found that the levels γ-H2AX, as an indicator of DNA DSBs, were markedly escalated in hepatomas formed in DEN-treated mice, and this was attenuated by SFN administration (Appendix A).

Satoh et al. reported that urethane-induced tumors were significantly smaller and less frequent in Keap1 knock-out mice than those in wild-type mice, suggesting that NRF2 intensifies host defense systems to prevent lung carcinogenesis; however, NRF2 activation after tumor initiation accelerates malignant cell growth [48]. Likewise, activation of NRF2 by SFN administration promoted the progression of pre-existing tumors in vinyl carbamate-induced lung carcinogenesis [22]. In addition, antioxidants *N*-acetylcysteine (NAC) and vitamin E markedly increased tumor progression and reduced survival in mouse models of B-Raf- and K-Ras-induced lung cancer [49]. Furthermore, NAC and a water-soluble vitamin E analog Trolox increased melanoma metastasis by inducing synthesis of reduced glutathione (GSH) in mice [50]. Notably, NAC and GSH promoted tumor formation and growth by reducing ROS and induction of TMBIM1 in DEN-induced murine hepatocarcinogenesis [51]. Consistent with these findings, our results lend support to the speculation that SFN-mediated NRF2 activation may accelerate the progression of pre-existing tumors in liver carcinogenesis. There is a limitation to the administration of SFN prior to carcinogen treatment in 15-day-old mice [52]. Thus, we could not assess the effect of SFN on the initiation of hepatocarcinogenesis in mice in the present study.

It was reported that *K-Ras*, *B-Raf*, and *Myc* oncogenes could promote the progression of pancreatic cancer via NRF2-mediated antioxidant signaling [53]. In addition, mutations of genes in the Ras/MAP kinase pathway drive DEN-induced liver tumorigenesis [4]. Somatic mutation of the *B-Raf* ^V600E^ has been found in HCC patients’ specimens [24,25]. In line with this notion, the present study revealed mutations of *B-Raf* ^V637E^ (the murine counterpart to human *B-Raf* ^V600E^) detected by Sanger DNA sequencing in the livers of DEN-treated mice. *B-Raf* mutation, together with mutations of Keap1 or NRF2, may accelerate HCC development. *NQO1*, one of NRF2 target genes, is predominantly overexpressed in HCC tumor tissues and hence considered a prognostic factor of HCC [54,55]. This suggests that aberrantly activated NRF2 exerts a pro-tumorigenic effect via NQO1 transcription in the promotion and progression of HCC. We speculate that mutation in *B-Raf* proto-oncogene may contribute, at least in part, to NRF2-mediated NQO1 overexpression to drive HCC development and that SFN treatment enhances oncogenic *B-Raf*-associated NRF2 activation, thereby up-regulating NQO1 expression to stimulate tumor growth.

SFN exhibits an anticancer effect in numerous cancer cell lines and animal tumor models through multiple mechanisms. Besides its direct effects on the transcription of genes involved in cancer cell signal transduction, epigenetic regulation is also of importance in the tumoricidal activity of SFN. For instance, SFN reduced the catalytic activity of DNA methyl transferases, histone deacetylases, and histone methyltransferases which accounted for the anticancer effect of SFN in HeLa cells [56]. However, in the tumor microenvironment, SFN blocks the T-cell-mediated immune response crucial for immune surveillance of tumors [57,58]. As SFN could act as a double-edged sword, it seems not advisable for a combination of SFN treatment with T-cell mediated cancer immunotherapy [59].

In summary, our present work shows that SFN alone failed to trigger tumor formation in mouse liver while it induces nuclear accumulation of NRF2. This suggests that without prior DNA damage and consequent oncogene/tumor suppressor gene mutation, NRF2 activation alone is not sufficient to induce hepatoma formation. In this context, SFN functions in the promotion and/or progression stage(s), rather than acting as an initiator causing gene mutation, in DEN-induced hepatocarcinogenesis (Figure 7). Nevertheless, the results are mostly derived from an animal model, and extrapolation to effects in humans remains speculative. Further, the physiologically achievable concentrations of this SFN as a tumor promotor vs. a chemopreventive principle of broccoli have not been defined yet. More clinically relevant studies are required to investigate the overall effects of SFN on human cancer development and progression.

## 4. Materials and Methods

### 4.1. Reagents and Antibodies

SFN was purchased from LKT Laboratories (St. Paul, MN, USA). DEN (Cat. N0258-1G) was purchased from Sigma-Aldrich (St. Louis, MO, USA). Antibodies for Keap1 (1:1000 for Western blotting), PCNA (1:1000 for Western blotting; 1:500 for immunohistochemistry), and NQO1 (1:1000 for Western blotting; 1:500 for immunohistochemistry and immunofluorescence staining) were products of Santa Cruz Biotechnology, Inc. (Dallas, TX, USA). NRF2 antibody (1:1000 for Western blotting; 1:200 for immunohistochemistry) was obtained from Abcam (Cambridge, UK). β-Actin (Santa Cruz, 1:1000), GAPDH (Santa Cruz, 1:1000), and Lamin B (Abcam, 1:1000) for Western blotting served as equal loading controls for whole-lysate, a cytosolic loading control, and a nuclear marker, respectively.

### 4.2. Animals

C57BL/6 mice were maintained in cages under controlled conditions. All animal experiments were approved by Seoul National University Animal Care and Use Committee (approval number: SNU-20140624-2-2).

### 4.3. DEN-Induced Hepatocellular Carcinoma Formation in Mice

Two-week-old male C57BL/6 mice received a single intraperitoneal dose of 25 mg/kg DEN prepared in PBS (pH 7.4). SFN suspended in PBS (pH 7.4) was daily injected intraperitoneally (25 mg/kg) for 3 months, starting at the 6th month after DEN administration (Figure 2A). Mice were sacrificed at the 9th month, and livers were isolated (*n* = 4, each group). To acquire aggressive tumors for sequencing, additional vehicle-treated and DEN-treated mice were sacrificed at the 12th month (*n* = 4, each group). Liver weight was measured, and visible nodules (≥0.5 mm diameter) were counted.

### 4.4. Tissue Lysis and Protein Extraction

Tissues were homogenized with lysis buffer [150 mM NaCl, 1% Triton × 100, 50 mM Tris-HCl (pH 7.4), 1 mM EDTA, 1 mM Na_3_VO_4_, 1 mM dithiothreitol (DTT), 0.1 mM PMSF and EDTA-free protease inhibitor cocktail tablets] for 2 h on ice followed by centrifugation at 12,000× *g* for 15 min. The protein concentration of the supernatant was determined by the Bio-Rad Bradford assay using bovine serum albumin (BSA) as a standard (Sigma-Aldrich, St. Louis, MO, USA).

### 4.5. Preparation of Cytosolic and Nuclear Extracts

Tissues were homogenized with hypotonic lysis buffer A [10 mM 4-(2-hydroxyethyl)-1-piperazineethanesulfonic acid (HEPES, pH 7.9), 1.5 mM MgCl_2_, 10 mM KCl, 0.5 mM DTT and 0.1 mM PMSF] for 2 h at 4 °C. The lysates were then mixed with 10% Nonidet P-40 (NP-40) for 10 min before centrifugation. After centrifugation at 12,000× *g* for 15 min, the supernatants were collected as cytosolic fractions. Remained pellets were washed three times with buffer A containing 10% NP-40 and resuspended in hypertonic lysis buffer C [20 mM HEPES (pH 7.9), 420 mM NaCl, 1.5 mM MgCl_2_, 20% glycerol, 0.2 mM EDTA, 0.5 mM DTT and 0.1 mM PMSF]. The nuclear lysates were vortex-mixed every 10 min for 2 h, followed by centrifugation at 12,000× *g* for 15 min. The supernatants were collected as nuclear extracts and subjected to measuring protein concentrations.

### 4.6. Western Blot Analysis

Protein lysates (20 μg) were subjected to 8–12% SDS-PAGE and transferred to the polyvinylidene difluoride (PVDF) membrane (Gelman Laboratory, Ann Arbor, MI, USA). The blots were blocked with 5% skim milk/TBST (Tris-buffered saline buffer containing 0.1% Tween-20) for 1 h at room temperature. The membranes were then incubated at 4 °C overnight with primary antibodies against Keap1, NRF2, NQO1, PCNA, GAPDH, Lamin B, and β-Actin. Blots were rinsed 3 times with TBST and then probed with peroxidase-conjugated secondary antibodies (Invitrogen, Carlsbad, CA, USA) for 1 h. Transferred proteins were detected with an enhanced chemiluminescence detection kit (Abclone, Seoul, Korea) and LAS-4000 image reader (Fujifilm, Tokyo, Japan).

### 4.7. Hematoxylin and Eosin (H&E) Staining

Livers were fixed in 4% paraformaldehyde (PFA) overnight, and the tissue block was embedded in paraffin. The paraffin-embedded 5 μm sections were stained with H&E. The slides were examined with a light microscope (Nikon, Tokyo, Japan).

### 4.8. Immunohistochemical Analysis

Livers were fixed in 4% PFA overnight, and the tissue block was embedded (5 μm sections) in paraffin. The sections were then deparaffinized in xylene and a series of graded (100%, 90%, 80%, 70%) alcohol baths. The sections were heated by microwave twice for 5 min each in 10 mM citrate buffer (pH 6.0) for antigen retrieval. Sections were treated with 3% H_2_O_2_ for 15 min to extinguish endogenous peroxidase activity. After a brief wash with PBS (pH 7.4), sections were blocked with 3% BSA in PBS for 1 h to diminish non-specific staining. Subsequently, the sections were incubated with primary antibodies with 1% BSA at 4 °C in a humidified chamber overnight. After being rinsed twice in PBS, the sections were incubated with a biotinylated peroxidase-conjugated secondary antibody (Vector Laboratories, Burlingame, CA, USA) for 1 h at room temperature. The sections were then incubated with avidin-peroxidase (Vector Laboratories, Burlingame, CA, USA) for 1 h. After rinsing 3 times in PBS, the sections were further stained by 3,3′-diaminobenzidine tetrahydrochloride solution and counterstained by Mayer’s hematoxylin. Stained slides were visualized under a light microscope (Nikon, Tokyo, Japan).

### 4.9. Immunofluorescence Staining

Liver specimens were fixed, paraffin-embedded, and sectioned, and the sections were then deparaffinized in xylene and a series of graded alcohols. The sections were heated by microwave twice for 5 min each in 10 mM citrate buffer (pH 6.0) for antigen retrieval. Sections were subjected to brief washing and permeabilized with 0.2% Triton X-100 for 1 h at room temperature. After a serial washing with PBS, the sections were blocked with 3% BSA in PBS for an additional 1 h. The sections were incubated with primary antibodies prepared in the blocking buffer at 4 °C in a humidified chamber overnight. On the next day, after being rinsed in PBS, the tissue sections were incubated with Alexa 488 and 546 secondary antibodies (Invitrogen, Carlsbad, CA, USA) for 1 h, followed by nuclear-counterstaining with DAPI. Immunofluorescence images were examined with a fluorescence microscope (Nikon, Tokyo, Japan).

### 4.10. Sanger Sequencing

Total RNA was extracted from vehicle- or DEN-treated liver tissue with DNeasy^®^ Blood & Tissue Kit (Qiagen, Hilden, Germany), and converted to cDNA using reverse transcriptase enzyme following the standard procedure. The cDNA region bearing murine *B-Raf* gene was amplified using PCR with HiPi Tag polymerase (ELPISbiotech, Daejeon, Korea). The PCR products were purified by QIAquick Purification Kit (Qiagen, Hilden, Germany). Sequencing was performed and cleaned up with BigDye^®^ Terminator v3.1 Cycle Sequencing Kit (Applied Biosystems, Waltham, MA, USA). Sequences were detected and analyzed by ABI PRISM 3730XL Analyzer (Applied Biosystems, Waltham, MA, USA).

### 4.11. Statistical Analysis

The statistical significance of differences between two groups was evaluated based on two-tailed Student’s *t* test. Statistical significance was accepted at *p* < 0.05 (* *p* < 0.05; ** *p* < 0.01; *** *p* < 0.001). Data were presented as mean ± SD. Data analyses were performed using GraphPad Prism 8.0 (GraphPad Software, San Diego, CA, USA).

## Figures and Tables

**Figure 1 ijms-23-05397-f001:**
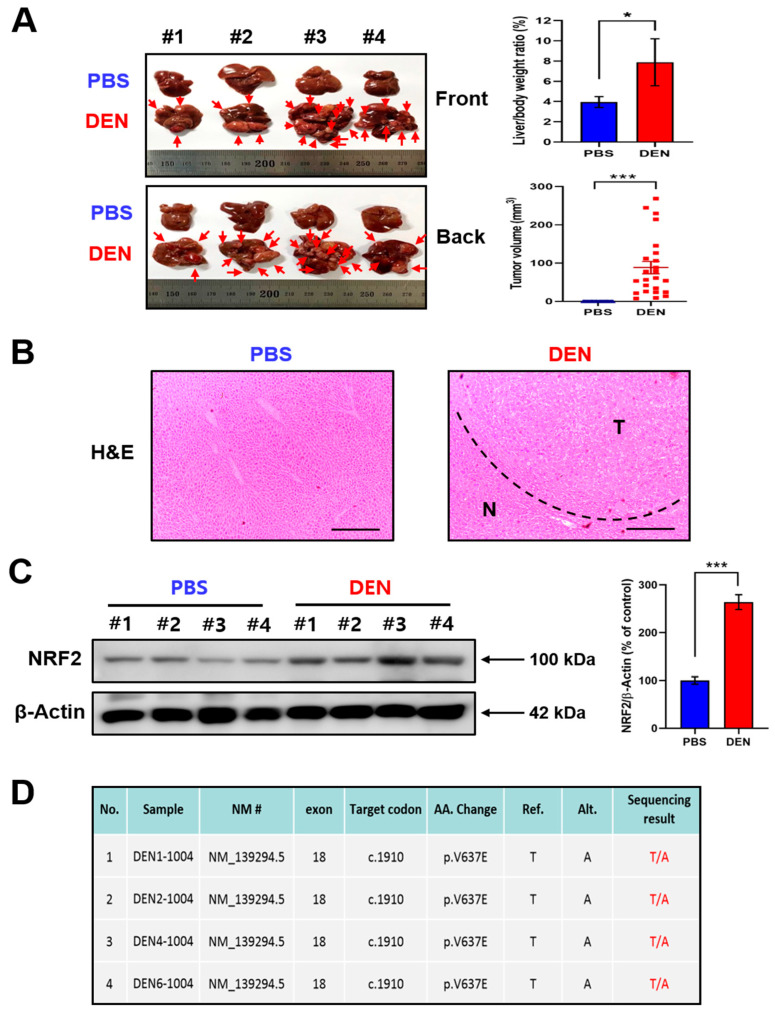
The elevated expression of NRF2 in DEN-induced hepatocarcinogenesis. (**A**) Two-week-old male mice (*n* = 4) were given a single intraperitoneal dose of DEN (25 mg/kg) prepared in PBS. Control animals received PBS alone. Twelve months after DEN administration, mice were sacrificed, and livers were isolated. Body weight, liver weight, and tumor volume were measured, and statistical significance was determined by the Student’s *t* test. Data are shown as the mean ± SD (*n* = 4). * *p* < 0.05; *** *p* < 0.001. (**B**) H&E staining was performed to determine the histologic difference in liver morphology between DEN-treated and vehicle-treated mice. N: normal tissue; T: tumor. Scale bar, 100 μm. (**C**) The expression of NRF2 in the livers of DEN- and vehicle-treated mice was measured by Western blot analysis. Statistical significance was determined by Student’s *t* test. Data are shown as the mean ± SD (*n* = 4). *** *p* < 0.001. (**D**) Total RNA samples were extracted from each group and converted to cDNA using reverse transcriptase following the standard procedure. After PCR amplification, Sanger sequencing was performed with BigDye^®^ Terminator v3.1 Cycle Sequencing Kit and analyzed by ABI PRISM 3730XL Analyzer.

**Figure 2 ijms-23-05397-f002:**
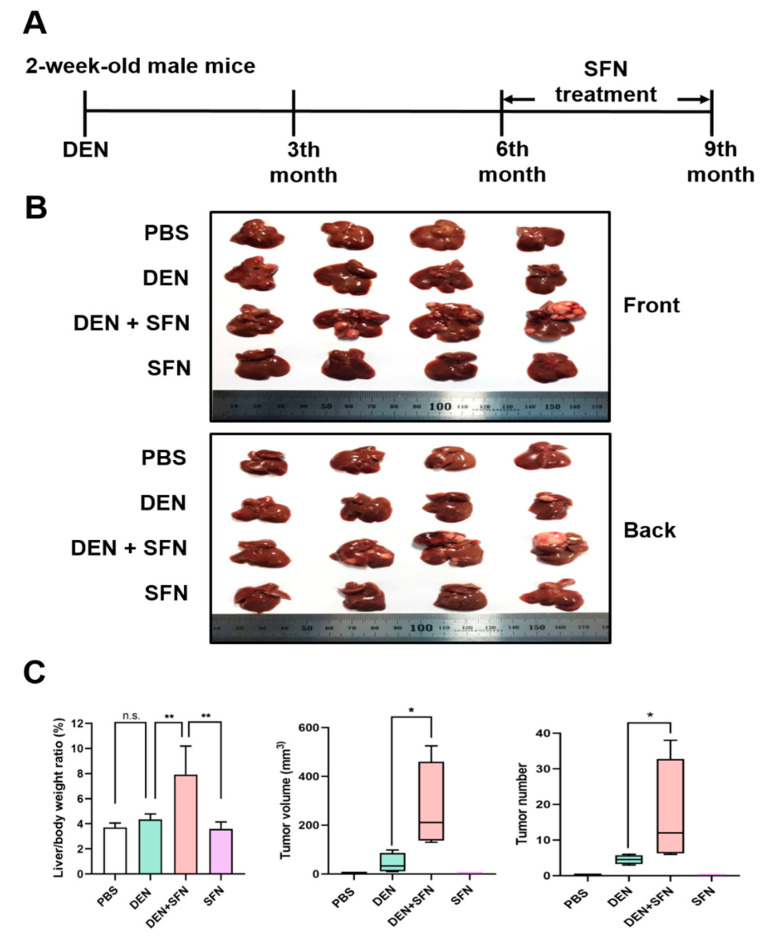
Effects of SFN on DEN-induced hepatocarcinogenesis. (**A**) Two-week-old male mice (*n* = 4) were given a single intraperitoneal dose of DEN (25 mg/kg) prepared in PBS. Six months after DEN administration, SFN was daily injected intraperitoneally (25 mg/kg) for 3 months. (**B**) At 9 months after DEN administration, mice were killed, and livers were isolated. (**C**) Body weight, liver weight, tumor volume, and tumor number were measured. The statistical significance was determined by the two-tailed unpaired Student’s *t* test. Data are shown as the mean ± SD (*n* = 4). * *p* < 0.05; ** *p* < 0.01; n.s.: non-significant.

**Figure 3 ijms-23-05397-f003:**
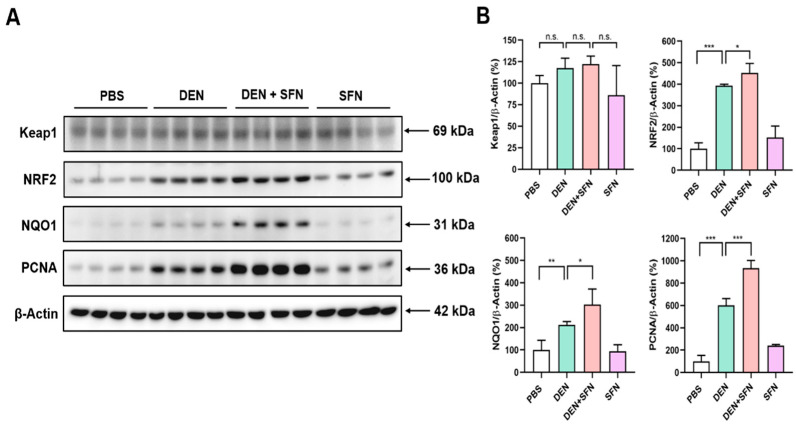
The effect of SFN on expression of NRF2, NQO1, and PCNA in the livers of DEN-treated mice. (**A**) Total protein lysates were prepared from livers of 9-month-old mice treated with vehicle (PBS), DEN, DEN plus SFN, or SFN alone (*n* = 4 for each group), and subjected to Western blotting with antibodies against Keap1, NRF2, NQO1, and PCNA. β-Actin served as an internal control. (**B**) The quantification of protein levels was conducted by using Image J software, and statistical significance was determined by the two-tailed unpaired Student’s *t* test. Data are shown as the mean ± SD. * *p* < 0.05; ** *p* < 0.01; *** *p* < 0.001; n.s.: non-significant.

**Figure 4 ijms-23-05397-f004:**
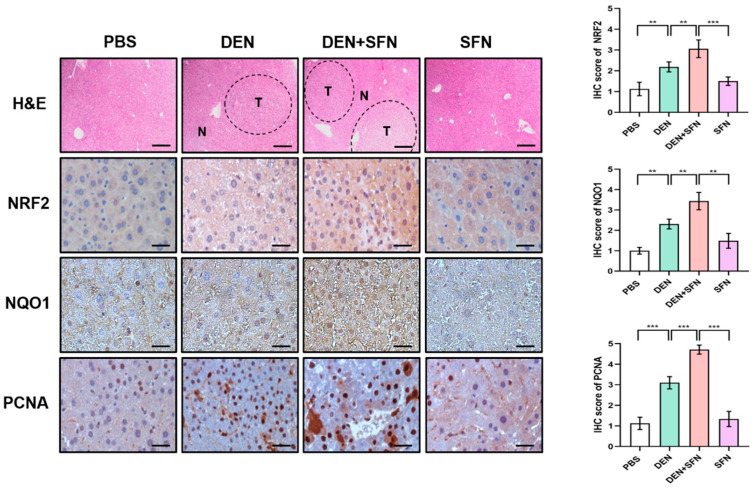
Immunohistochemical (IHC) analysis of NRF2, NQO1, and PCNA expressed in DEN-induced murine hepatocarcinogenesis with and without SFN administration. The paraffin sections of liver tissues were subjected to H&E staining for histopathological evaluation. N: normal tissue; T: tumor. Scale bar, 500 μm. The paraffin sections of liver tissues were subjected to IHC staining with antibodies against NRF2, NQO1, and PCNA. The IHC score was analyzed by the image processing program Image J, and results are shown as the mean ± SD of 4 samples for each group. ** *p* < 0.01; *** *p* < 0.001. Representative images of stained sections are displayed. Scale bar, 100 μm.

**Figure 5 ijms-23-05397-f005:**
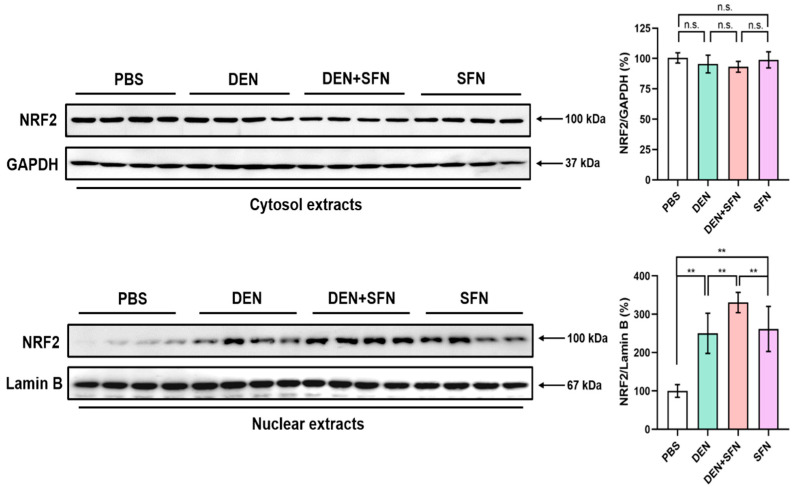
Effect of SFN administration on the nuclear translocation of NRF2 in the livers of DEN-treated mice. Cytosol and nuclear extracts prepared from the livers of 9-month-old mice treated with vehicle (PBS), DEN, DEN plus SFN, or SFN alone (*n* = 4 for each group) were subjected to Western blotting with antibodies against NRF2. GAPDH and Lamin B served as internal controls for cytosol and nuclear extracts, respectively. The quantification of protein levels of NRF2 was conducted, and the statistical significance was determined by the two-tailed unpaired Student’s *t* test. Data are shown as the mean ± SD. ** *p* < 0.01; n.s.: non-significant.

**Figure 6 ijms-23-05397-f006:**
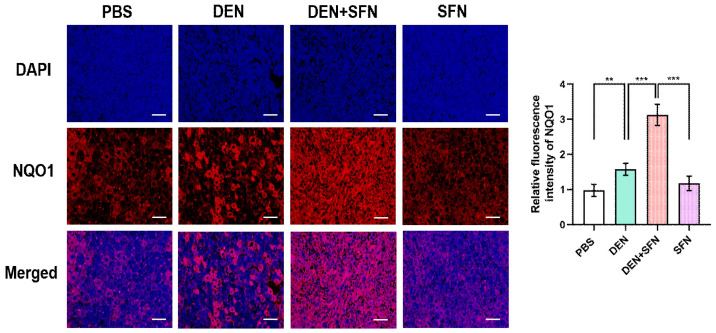
Immunofluorescence staining of NQO1 in the livers of control mice, DEN-treated mice with and without SFN administration, and mice treated with SFN alone. The paraffin sections of liver tissues were subjected to immunofluorescence staining with antibodies against NQO1 (red). DAPI (blue) was used to label cellular nuclei. Images were visualized under a fluorescence microscope. The fluorescent intensity was analyzed by the image processing program Image J and results are shown as the mean ± SD of 4 samples for each group. ** *p* < 0.01; *** *p* < 0.001. Representative images of stained sections are displayed. Scale bar, 100 μm.

**Figure 7 ijms-23-05397-f007:**
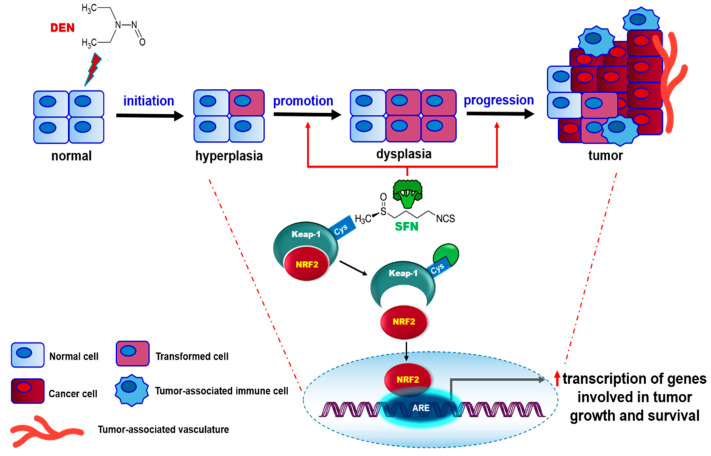
A proposed scheme illustrating the role of SFN in DEN-induced hepatocarcinogenesis.

## Data Availability

Data presented in this study are included in the article and its Appendix A.

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
