# Peer review of "Tumor Promoting Effects of Sulforaphane on Diethylnitrosamine-Induced Murine Hepatocarcinogenesis"

_ijms, 2022, doi:10.3390/ijms23105397_

Round 1

Reviewer 1 Report

In this article the authors assessed in the animal model (mice), the tumor promoting effects of sulforaphane on diethylnitrosa- 2 mine-induced murine hepatocarcinogenesis.

The rational of this study as well as its methods are appropriate.

I suggest that the last sentence of the section Discussions (lines 373-375) should be deleted and replaced with a sentence highlighting that these data on animal model, currently, cannot be translated in humans but further studies are required to investigate if these findings could involve humans.

 Considering these interesting findings, the authors should comment in the section discussion how they could explain the anticancer effect of sulforaphane on HeLa cells reported by Kedhari Sundaram M et al. Minerva Medica 2021;112:792-803 Is it only a question of model used ?

Author Response

Reviewer #1
In this article the authors assessed in the animal model (mice), the tumor promoting

effects of sulforaphane on diethylnitrosa
- 2 mine-induced murine
hepatocarcinogenesis.

The rational of this study as well as its methods are appropriate.

I suggest that the last sentence of the section Discussions (lines 373
-375) should be
deleted and replaced with a sentence highlighting that these data on animal model,

currently, cannot be
translated in humans but further studies are required to
investigate if these findings could involve humans.

Author response:
In accordance with the reviewers valuable suggestion, we deleted
the
following 3 sentences in the last paragraph in the Discussion section; Results of
the present study suggest that the “dark side” of SFN outweighs its benefits. Thus, it

remains controversial whether SFN is good or not in cancer management.

Identification of the turning point at which NRF2 changes from bei
ng a tumor
suppressor to a tumor promoter and the differential role of NRF2 in tumor

microenvironment merit further investigation
. These deleted sentences have been
replaced by
: “In summary, our present work showed that SFN alone failed to trigger
tumor f
ormation in mouse liver while it induces nuclear accumulation of NRF2. This
suggests that NRF2 activation alone is not sufficient to induces hepatoma formation.

In this context, SFN may function as a tumor promoter, rather than as an initiator

causing gene
mutation, in DEN-induced hepatocarcinogenesis. Nevertherless, the
results are mostly derived from an animal model, and extrapolation to effects in

humans remains speculative. Further, the physiologically achievable concentrations

of this phytochemical as
a tumor promotor vs. a chemopreventive principle of broccoli
have not been defined yet. More clinically relevant studies are required to investigate

the overall effects of SFN on human cancer development and progression”.

Considering these interesting fin
dings, the authors should comment in the section
discussion how they could explain the anticancer effect of sulforaphane on HeLa cells

reported by Kedhari Sundaram M et al. Minerva Medica 2021;112:792
-803 Is it only a
question of model used?

Author response: Thanks for introducing the work by Kedhari Sundaram et al. on
anti
-cancer effects exerted by SFN via epigenetic modulation of genes in HeLa cells.
In accordance with the reviewer
s suggestion, we highlighted the anticancer effects of
SFN via epig
enetic regulation by citing this important publication as a new re

Reviewer 2 Report

The study of Zheng et al. described tumor promotive effects of sulforaphane (SFN) in diethylnitrosamine-treated mouse livers. It was assumed by the authors that this is mediated by the nuclear factor erythroid 2-related factor 2 (NRF2). Moreover, the authors suggest that this involves an oncogenic BRAF mutation. There are several aspects, that should be addressed by the authors to further improve their manuscript:

The novelty of this study remains unclear to the reader and should be outlined by the authors, since related studies already exist that describe tumor promotive effects of SFN via NRF2. In addition, many statements made by the authors have not been supported by suitable experiments. I would recommend focusing on findings that can be drawn from this study and to clearly signify hypotheses and assumptions to the reader.

Figure 2: The liver to body weight ratio is more informative and commonly used.

Line 127: The abbreviations NQO1 and PCNA should be explained in the main text when mentioned first.

Molecular weights of protein bands are provided in the original blots but not in the figures 1 and 3 of the main text.

Figure 4: The number of histologic analyses or independent experiments remains unclear. Please mention this in the figure legend.

Line 144: “Enhanced nuclear translocation” or just a general increase in NRF2 protein amount? Please note that to measure the translocation of the transcription factor, the amount of NRF2 in the cytosol and nucleus must be compared in a temporal context.

Fig. 5A: SFN treatment alone already increased nuclear levels of NRF2 but this was not mentioned or discussed by the authors.

Fig. 5B: The fluorescence images are very small, show weak fluorescence and no visible nuclear staining of NRF2.The NRF2 staining in Fig. 4 is much better. It is unclear to me why the authors include fluorescence images of NRF2.

Fig. 6: The immunofluorescence of NQO1 is barely visible and requires improvement (see Fig. 5B).

Line 212: “…appeared to protect tumors from oxidative DNA damage,” This was not investigated by the authors in the present study. Protective effects of NQO1 have been reported by others.

Line 246: SFN alone is triggered no tumor formation in the present study. So, a second hit is required (e.g. mutation?) and this should be made clear by the authors in the section “Discussion”.

Lines 369-375: It is better to delete the “Conclusion” section, as none of the statements which are given here have been carefully investigated by the authors in the present study.

Author Response

Reviewer #2
The study of Zheng et al. described tumor promotive effects of sulforaphane (SFN) in

diethylnitrosamine
-treated mouse livers. It was assumed by the authors that this is
mediated by the nuclear factor
erythroid 2-related factor 2 (NRF2). Moreover, the
authors suggest that this involves an oncogenic BRAF mutation. There are several

aspects, that should be addressed by the authors to f
urther improve their manuscript
1.
The novelty of this study remains unclear to the reader and should be outlined by
the authors, since related studies already exist that describe tumor promotive effects

of SFN via NRF2. In addition, many statements made by the authors have not been

supported by suitable experiments. I would recommend focusing on findings that can

be drawn from this study and to clearly signify hypotheses and assumptions to the

reader.

Author response:
Aberrant activation of NRF2 in various human malignancies as well
as experimentally induced
neoplasm has been frequently reported and reviewed within
recen
t years. According to the database provided by TNMplot [1], even in lung cancer,
differential
patterns of NRF2 expression levels were observed in lung adenocarcinoma
and lung small
-cell carcinoma. However, differential roles of NRF2 in multi-stage
carcinogenesis and diverse
circumstances in tumor microenvironment remain largely
un
resolved.
Considering the recently addressed oncogenic potential of NRF2, tumor promoting

effect of some NRF2 activators including SFN has been speculated by researchers in

this area. In line with this clinical observation, the
pro-oncogenic effect of SFN has been reported in experimentally induced lung cancer. Our previous study
demonstrated the role of NRF2 in DEN
-induced murine hepatocarcinogenesis [2], and
this prompted us to assess the effects of SFN administered during post
-initiation
period in the same animal model.

2.
Figure 2: The liver to body weight ratio is more informative and commonly used.
Author response:
The authors acknowledge the reviewer’s constructive comments.
We
accordingly analyzed the liver to body weight ratio and added the corresponding
bar graph in Figure
1A and Figure 2C. These updated quantifications are included in the revised manuscript.

3. Line 127: The abbreviations NQO1 and PCNA should be explained in the main text
when mentioned first.

Author response:
The authors apologize for the carelessness. The abbreviations of
NQO1
(NAD(P)H: quinone oxidoreductase 1) and PCNA (proliferating cell nuclear
antigen)
are now defined in the first use (line131-132).

4. Molecular weights of protein bands are provided in the original blots but not in the
figures 1 and 3 of the main text.

Author response:
In accordance with the reviewer’s suggestion, molecular weights
of protein bands are added in the main
Figures 1C, 3A, and 5 of the revised manuscript.
5.
Figure 4: The number of histologic analyses or independent experiments remains
unclear. Please mention this in the figure le
gend.
Author response:
The statement, Representative images of stained sections are
displayed (
n=4 for each group).” was added in figure legend, line 157.

6. Line 144: “Enhanced nuclear translocation” or just a general increase in NRF2
protein amount?
Please note that to measure the translocation of the transcription
factor, the amount of NRF2 in the cytosol and nucleus must be compared in a temporal

context.

Author response:
The authors acknowledge the reviewer’s valuable comments. We
measured
the cytosolic expression of NRF2 in mouse hepatomas of and added the
data in Figure 5. Corresponding quantification data are also displayed. Uncropped

image
s of cytosolic NRF2 and α-Tubulin are supplied in the supplementary data.
Explanations of data were
added and revised in the Result section: “Nuclear
expression of NR
F2 was markedly elevated in the livers of DEN-treated mice
compared with
that in PBS-treated mice. The migration of NRF2 to nucleus was further
enhanced by SFN administration
in DEN-treated mice. The increased nuclear protein
expression of NRF2 was associated with decreased cytosolic NRF2 expression in

SFN plus DEN
-treated mice compared with SFN alone-treated mice. However, there
was no
substantial difference in cytosolic NRF2 expression between PBS-treated and
DEN
-treated mice.”

7. Fig. 5A: SFN treatment alone already increased nuclear levels of NRF2 but this was
not mentioned or discussed by the authors.

Author response:
In recognition of the reviewer’s concern, we added this information
in
the Result section of the revised manuscript.

8. Fig. 5B: The fluorescence images are very small, show weak fluorescence and no
visible nuclear staining of NRF2.The NRF2 staining in Fig. 4 is much better. It is

unclear
to me why the authors include fluorescence images of NRF2.
Author response:
Thanks for the comments. The immunofluorescence image of
NRF2 has been
replaced by a better one in Fig.6A.
9.
Fig. 6: The immunofluorescence of NQO1 is barely visible and requires
improvement (see Fig. 5B).

Author response:
Thanks for the comments. The immunofluorescence images of
NQO1
are replaced with better ones (Fig. 6B).

10. Line 212: “...appeared to protect tumors from oxidative DNA damage,” This was
not investigated by the
authors in the present study. Protective effects of NQO1 have
been reported by other
s.
Author response:
DNA damage sensor γ-H2AX was increased in preneoplastic
lesions of hepatocellular carcinoma
, and might be a useful biomarker for predicting the
risk of
HCC [3]. We detected the expression of γ-H2AX protein with
immunohistochemical
analysis. We found that the expression level of γ-H2AX was
higher in DEN
-induced hepatomas compared with that of normal livers in vehicle
(PBS)
-treated mice, and SFN treatment attenuated it in DEN-treated mice. In addition,
SFN treatment alone did not cause any DNA damage in mouse livers.
These data are
provided as Suppleme
ntary Fig. 3 in the revised manuscript.

11. Line 246: SFN alone is triggered no tumor formation in the present study. So, a
second hit is required (e.g. mutation?) and this should be made clear by the authors

in the section “Discussion”.

Author response:
As noted by the reviewer, SFN alone failed to trigger tumor
formation. This suggests that SFN may function as a tumor promoter, rather than as

a
n initiator causing gene mutation, in DEN-induced hepatocarcinogenesis. This issue
is addressed in the Discussio
n section.

12. Lines 369-375: It is better to delete the “Conclusion” section, as none of the
statements which are given here have been carefully investigated by the authors in

the present study.

Author response:
In accordance with the reviewer’s suggestion, we have deleted the
Conclusion section in the revised version of the manuscript.

Bibliography

1.
Bartha, A.; Gyorffy, B. TNMplot.com: A web tool for the comparison of
gene expression in
normal, tumor and metastatic tissues.
Int J Mol Sci

2021
,
22
, doi:10.3390/ijms22052622.
2.
Ngo, H.K.C.; Kim, D.H.; Cha, Y.N.; Na, H.K.; Surh, Y.J. Nrf2 mutagenic
activation drives h
epatocarcinogenesis.
Cancer Res
2017,
77
, 4797-4808,
doi:10.1158/0008
-5472.CAN-16-3538.
3.
Matsuda, Y.; Wakai, T.; Kubota, M.; Osawa, M.; Takamura, M.; Yamagiwa,
S.; Aoyagi, Y.; Sanpei, A.; Fujimaki, S. DNA damage sensor gamma
-H2AX
is increased in preneoplastic lesions of hepatocellular carcinoma.

ScientificWorldJournal
2013,
20
13, 597095, doi:10.1155/2013/597095.

Round 2

Reviewer 2 Report

Zheng et al. improved their manuscript that described tumor promotive effects of sulforaphane (SFN) in diethylnitrosamine-treated mouse livers. However, there are still some aspects that require further improvement:

Line 65-70: Related studies describing tumor promotive effects of SFN via NRF2 already exist. These studies should be mentioned in the section “Introduction”. Moreover, explain to the reader novel aspects of your study.

Figure 5: The alpha-tubulin protein levels decrease in the SFN-treated group. SFN has been described to induce cytoskeletal reorganization which may explain the decrease of alpha-tubulin in your experiments. Therefore, other proteins such as GAPDH should be selected as loading controls for Western blotting.

Figure 5, legend: The cytosolic NRF2 analyses remained unaddressed in the figure legend.

Line 163: Nuclear NRF2 protein levels or … protein amounts instead of nuclear expression. The term expression is commonly used when gene expression is meant.

Figure 6A/B: The immunofluorescence images of NRF2 and NQO1 have been improved. However, in contrast to Figure 4, most cells seem to have no nuclear staining of NRF2. This requires explanation. You may also quantify the intensity of the immunofluorescence for both proteins using appropriate software.

Supplementary Figure 3: No quantitative data have been provided by the authors. In addition, the number of experiments is missing in the figure legend. In addition, other groups reported increased DNA double strand breaks in cancer cells by SFN (PMID: 18854174). This aspect should be discussed.

Line 292: “…to induce…”

Line 297: You should explain to the reader that broccoli contains SFN. This is not necessarily known.

Author Response

Zheng et al. improved their manuscript that described tumor promotive effects of
sulforaphane (SFN) in diethylnitrosamine
-treated mouse livers. However, there are
still some aspects that require further improvement:

1.
Line 65-70: Related studies describing tumor promotive effects of SFN via NRF2
already exist. These studies should be mentioned in the section “Introduction”.

Moreove
r, explain to the reader novel aspects of your study.
Author reply:
In accordance with the reviewer’s valuable suggestion, we added the
following statements to the
last paragraph of INTRODUCTION: The differential effects
of NRF2 on multi
-stage carcinogenesis have raised a concern about the validity of
NRF2 inducers/activators for chemoprevention.
Sulforaphane (SFN) has been known
to be a powerful inducer/activator of NRF2.
The majority of health beneficial effects of
broccoli consumption have been considered to be mediated by SFN. However, SFN

is present in broccoli and other cruciferous vegetables as its precursor, glucoraphanin,

SFN is formed through the actions of myrosinase, a β
-thioglucosidase present in either
the plant tissue or the mammalian microbiome
[1]. Despite well-known
chemopreventive potential
of SFN, pro-oncogenic effects of this prototypic
phytochemical with pr
onounced NRF2 inducing activity have been reported in lung
cancer and colorectal cancer
[2,3]. However, the effects of SFN given during post-
initiation period of tumorigenesis remain unresolved.
In the previous study, we
demonstrated the role of NRF2 in chemically
-induced liver cancer development in
mice
[4]. This prompted us to assess the effect of SFN on DEN-induced murine
hepatocarcinogenesis.

2. Figure 5: The alpha-tubulin protein levels decrease in the SFN-treated group. SFN
has been described to induce
cytoskeletal reorganization which may explain the
decrease of alpha
-tubulin in your experiments. Therefore, other proteins such as
GAPDH should be selected as loading controls for Western blotting.

Author reply:
We agree with the reviewer’s speculation that the decrease of alpha-
tubulin in
our experiments might be attributable to cytoskeletal reorganization induced
by SFN
. To avoid this complication, we re-probed the membrane previously used for
the measurement of
the NRF2 expression with stripping buffer, and detected the
GAPDH protein expression in the same membrane.
The updated images and
quantification data
are shown in new Figure 5. We also added the information on the
use of
GAPDH as an equal loading control in the corresponding “Figure legend” and
the
“Materials and Methods” section.

3. Figure 5, legend: The cytosolic NRF2 analyses remained unaddressed in the figure
legend.

Author reply:
The authors apologize for the carelessness. We added information
about cytosolic NRF2 analyses
in the legend to Figure 5 of the revised manuscript.

4. Line 163: Nuclear NRF2 protein levels or ... protein amounts instead of nuclear
expression. The term expression is commonly used when gene expression is
meant.
Author reply:
In recognition of the reviewer’s concern, we corrected this information
in the
Result section of the revised manuscript.
5.
Figure 6A/B: The immunofluorescence images of NRF2 and NQO1 have been
improved. However, in contrast to
Figure 4, most cells seem to have no nuclear
staining of NRF2. This requires explanation. You may also quantify the intensity of the

immunofluorescence for both proteins using appropriate software.

Author reply:
As the quality of immunofluorescence staining of nuclear NRF2 is poor,
we have deleted the
corresponding immunofluorescence image. In addition, we
quantifi
ed the immunofluorescence image of NQO1 and added the quantification data
in Fig.
6. The fluorescent intensity was analyzed by the image processing program
Image J and results are shown as the mean
± S.D. of 4 samples for each group. We
also revised the sentences accordingly in
the “Result” section and the corresponding
figure legend
.

6. Supplementary Figure 3: No quantitative data have been provided by the authors.
In addition, the number of experiments is missing in the figure legend. In addition,

other groups reported increased DNA double strand breaks in cancer cells by SFN

(P
MID: 18854174). This aspect should be discussed.
Author reply:
Thanks for the constructive comments. The sentences The IHC score
was analyzed by the image processing program Image J and results are shown as the

mean ± S.D. of 4 samples for each group. ***
p<0.001; n.s.: non-significant.
Representative images of stained sections are displayed.”
are now included in the
legend of Supplementary Figure 3.
In addition, we quantified the immunohistochemical
images of gamma
-H2AX and added the quantification data. Meanwhile, other groups
reported SFN induced DNA double strand breaks in cancer cells, we cited this

important reference and
added discussion in the corresponding section, “It was

reported that SFN induced DNA double strand breaks (DSBs) in HeLa cells,
subsequently
resulting in cancer cell death [5]. However, other studies demonstrated
that
SFN could repair DSB-induced DNA damage [2,6,7]. Both normal tissues and
tumor tissues may benefit from such reparative effect
s of SFN, resulting in tumor
preventive and tumor promotive effects, respectively. In accordance with this notion,

we found that the expression of
γ-H2AX was markedly escalated in hepatomas
formed in DEN
-treated mice, and this was attenuated by SFN administration
(Supplementa
ry Fig. 3).”
In addition, we also added quantification data of IHC images of NRF2, NQ
O1 and
PCNA
in the main Figure 4, and the quantification information is included in the legend.

7. Line 292: “...to induce...”
Author reply:
The authors apologize for the carelessness. We corrected the
grammatical error in the corresponding section.

8
. Line 297: You should explain to the reader that broccoli contains SFN. This is not
necessarily known.

Author reply: In accordance with the reviewers valuable suggestion, we added the
correct information on broccoli-derived SFN in the Introduction section; The
majority of health beneficial effects of broccoli consumption have been considered to

be mediated by SFN. However, SFN is present in broccoli and other cruciferous
vegetables as its precursor, glucoraphanin, SFN is formed through the actions of
myrosinase, a β-thioglucosidase present in either the plant tissue or the mammalian
microbiome [1].

9. Finally, we have prepared a summary figure that we would like to include in the
revised version of the manuscript as Figure 7 with a legend, “A proposed scheme
illustrating the role of SFN in DEN-induced hepatocarcinogenesis”. We hope that this
will help readers better understand the work addressed in the manuscript.

1. Yagishita, Y.; Fahey, J.W.; Dinkova-Kostova, A.T.; Kensler, T.W. Broccoli
or Sulforaphane: Is It the Source or Dose That Matters?

Molecules
2019,
24
, doi:10.3390/molecules24193593.
2.
Tao, S.; Rojo de la Vega, M.; Chapman, E.; Ooi, A.; Zhang, D.D. The
effe
cts of NRF2 modulation on the initiation and progression of chemically
and genetically induced lung cancer.

Mol Carcinog
2018,
57
, 182-192,
doi:10.1002/mc.22745.

3.
Gwon, Y.; Oh, J.; Kim, J.S. Sulforaphane induces colorectal cancer cell
proliferation throu
gh Nrf2 activation in a p53-dependent manner.
Appl Biol

Chem
2020,
63
, doi:ARTN 86
10.1186/s13765
-020-00578-y.
4.
Ngo, H.K.C.; Kim, D.H.; Cha, Y.N.; Na, H.K.; Surh, Y.J. Nrf2 Mutagenic
Activation Drives Hepatocarcinogenesis.

Cancer Res
2017,
77
, 4797-4808,
doi:10.1158/0008
-5472.CAN-16-3538.
5.
Sekine-Suzuki, E.; Yu, D.; Kubota, N.; Okayasu, R.; Anzai, K. Sulforaphane
induces DNA double strand breaks predominantly repaired by homologous

recombination pathway in human cancer cells.

Biochem Biophys Res

Commun
2008,
377
, 341-345, doi:10.1016/j.bbrc.2008.09.150.
6.
Harris, C.M.; Zamperoni, K.E.; Sernoskie, S.C.; Chow, N.S.M.; Massey, T.E.
Effects of in vivo treatment of mice with sulforaphane on repair of DNA

pyridyloxylbutylation.

Toxicology
2021,
454
, 152753,
d
oi:10.1016/j.tox.2021.152753.
7.
Ding, Y.; Paonessa, J.D.; Randall, K.L.; Argoti, D.; Chen, L.; Vouros, P.;
Zhang, Y. Sulforaphane inhibits 4
-aminobiphenyl-induced DNA damage in
bladder cells and tissues.

Carcinogenesis
2010,
31
, 1999-2003,
doi:10.1093/car
cin/bgq183

Round 3

Reviewer 2 Report

Minor comments:

Line 235: “…SFN could repair DSB-induced DNA damage…” This is certainly not the case. Protein complexes are involved in DSB repair. Moreover, DSB are not the reason for DNA damage, but its expression. "SFN seem to promote DSB repair" might be better option.

Line 238: “…expression of gamma-H2AX” The abundance of gamma-H2AX as an indicator of DSB increased. The term “expression” is not correct here.

Author Response

1. Line 235: “...SFN could repair DSB-induced DNA damage...” This is certainly not
the case. Protein complexes are involved in DSB
repair. Moreover, DSB are not the
reason for DNA damage, but its expression. "SFN seem to promote DSB repair"

might be better option.

Author reply:
The authors acknowledge the reviewers concern about the statement
on
SFN could repair DSB-induced DNA damage and agree with the corrections
suggested
. In accordance with the reviewer’s valuable suggestion, the paragraph
descri
bing the effects of SFN on the DNA damage response in terms of the levels of
γ
-H2AX in the hepatoma is rewritten more precisely with additional citation of a
relevant reference
[44]: ROS at low levels may cause mutations and genomic
instability through DNA double strand breaks (DSBs)
[44]. DSBs can be lethal to a cell,
and cause apoptosis if not repaired. It was reported that SFN induced DSBs in HeLa

cells, subsequently resulting in cancer cell death
[45]. However, other studies have
demonstrated that
SFN seems to protect against DNA damage [46,47]. Both normal
tissues and tumor tissues may benefit from such protective effect of SFN, resulting in

tumor preventive and tumor promotive effects, respectively
”.
44
. Sharma, V.; Collins, L.B.; Chen, T.H.; Herr, N.; Takeda, S.; Sun, W.; Swenberg, J.A.; Nakamura, J.
Oxidative stress at low levels can induce clustered DNA lesions leading to NHEJ mediated

mutations.

Oncotarget
2016,
7
, 25377-25390, doi:10.18632/oncotarget.8298.
45
. Sekine-Suzuki, E.; Yu, D.; Kubota, N.; Okayasu, R.; Anzai, K. Sulforaphane induces DNA double
strand breaks predominantly repaired by homologous recombination pathway in human

cancer cells.

Biochem Biophys Res Commun
2008,
377
, 341-345,
doi:10.101
6/j.bbrc.2008.09.150.
46
. Ding, Y.; Paonessa, J.D.; Randall, K.L.; Argoti, D.; Chen, L.; Vouros, P.; Zhang, Y. Sulforaphane
inhibits 4
-aminobiphenyl-induced DNA damage in bladder cells and tissues.
Carcinogenesis

2010
,
31
, 1999-2003, doi:10.1093/carcin/bgq183.
4
7. Harris, C.M.; Zamperoni, K.E.; Sernoskie, S.C.; Chow, N.S.M.; Massey, T.E. Effects of in vivo
treatment of mice with sulforaphane on repair of DNA pyridyloxylbutylation.

Toxicology

2021
,
454
, 152753, doi:10.1016/j.tox.2021.152753.

2. Line 238: “...expression of gamma-H2AX” The abundance of gamma-H2AX as an
indicator of DSB increased. The term “expression” is not correct here.

Author reply:
The authors acknowledge the reviewers valuable suggestion. The term
expression was corrected accordingly by replacing it with levels in the corresponding
se
ction of the revised manuscript: In line with this notion, we found that the levels of
γ
-H2AX, as an indicator of DSB, were markedly escalated in hepatomas formed in
DEN-treated mice ...”.

3.
Finally, the authors have made additional editorial changes for linguistic
improvement. These include:

1)
L8 in the first paragraph under Introduction: induce liver tumors  induce liver
tumor formation

2)
L45-46: host defense mechanisms  host anticancer defense mechanisms
3) L47:
promote the survival and progression of cancer cells  promote the survival
and growth of cancer cells

4)
L68-72: However, SFN is present in broccoli and other cruciferous vegetables as
its precursor, glucoraphanin,
SFN is formed through the actions of myrosinase, a β-
thioglucosidase present in either the plant tissue or the
mammalian microbiome [21].

However, SFN is present in broccoli and other cruciferous vegetables as its
precursor, glucoraphanin,
and is released through the action of myrosinase, a β-
thioglucosidase present in either the plant tissue or the
gut microbiome after
i
ntensive chewing or during digestion, respectively [21].
5) L75: remain unresolved
 remain poorly resolved
6) L93: with poor
prognosis/survival and increased risk of  with poor prognosis and
increased risk of

7) L97-99:
capillary electropherogram (Supplementary Fig. 1B)., and 4 cases of B-Raf
V367E
mutation (n=7) were detected in DEN-induced liver tumors (Fig. 1D).  capillary

electropherogram (Supplementary Fig. 1B). Of the 7 hepatoma samples from DEN-
treated mice, 4 cases of B-Raf V367E mutation (n=7) were detected (Fig. 1D).

8) L99-100: and detailed information of all specimens was listed in
 and detailed
information of all specimens is listed in

9) L120-121: 2.3. SFN Treatment Promoted Tumor Growth in a DEN-induced Mouse
Model  2.3. SFN Treatment Promoted Tumor Growth in a DEN-induced Murine
Hepatocarcinogenesis Model

10) L129-130: 2.4. The Expression Levels of NRF2, NAD(P)H: quinone
oxidoreductase 1 (NQO1), and proliferating cell nuclear antigen (PCNA) Were Higher
in the SFN-treated Group Than the Vehicle-treated Group in DEN-treated Mice  2.4.
The Expression Levels of NRF2, NAD(P)H: quinone oxidoreductase 1 (NQO1), and
Proliferating Cell Nuclear Antigen (PCNA) Were Further Elevated by SFN
Administration in DEN-treated Mice

11) L132-133:
SFN treatment further enhanced the expression level of NRF2, N
QO1
, and PCNA in DEN-treated mice (Fig. 3).  SFN treatment further enhan
ced the expression level
s of NRF2, NQO1, and PCNA in DEN-treated mice (Fi
g. 3).

12) L133-134: However, there was no significant differences the expression of
Keap1  However, there was no significant differences in the expression level
s of Keap1.

13) L187-188: DEN-treated mice compared with those in PBS-treated mice.-->
DEN-treated mice compared with those in PBS-treated control mice.

14) L190-191: protein levels of NRF2 between DEN-treated mice and vehicle-tr
eated mice. (Fig. 5).--> protein levels of NRF2 between DEN- and vehicle-treat
ed mice. (Fig. 5).

15) L213-215: Numerous studies reported that SFN exhibits chemopreventive ef
fects in various tumor types via through induction of  Numerous studies have
reported that SFN exhibits exerts chemopreventive effects in various tumor type
s models via induction/activation of

16) L225-226: Collapse of NRF2-mediated antioxidant signaling leads to elevate
d ROS levels  Collapse of NRF2-mediated antioxidant signaling gives rise to
elevated ROS levels

17) L229: is anticipated to protect tumors from
 is hence anticipated to rescue
cancer cells from

18) L230: promoted tumor growth
 promotes tumor growth
19) L231-232: The DNA damage sensor γ-H2AX was increased in preneoplastic
lesions of HCC
 The DNA damage sensor γ-H2AX was found to be
overexpressed in preneoplastic lesions of HCC

20) L243: Keap1-knock-out
 Keap1 knock out
21) L256:
Thus, we could not evaluate assess the effect of SFN  Thus, we could
not assess the effect of SFN

22) L258: It was reported that K-Ras, B-Raf, and Myc oncogenes
 It was reported
that K-Ras, B-Raf, and Myc oncogenes

23) L261: has been found in human HCC patients specimens
 has been found in
HCC patients specimens

24) L261-262: In agreement with this notion  In line with this notion

25) L264-265: Such mutation of B-Raf oncogene-induced NRF2 activation, toget
her with genetic mutations of Keap1 or NRF2, may contribute to aberrant act
ivation of NRF2 in HCC development  B-Raf mutation, together with mutati
ons of Keap1 or NRF2, may accelerate HCC development

26) L268-269: . We speculate that mutations in B-Raf proto-oncogene contribut
e, at least in part, to activation of NRF2-mediated NQO1 overexpression 

We speculate that mutation in B-Raf proto-oncogene may contribute, at least
in part, to NRF2-mediated NQO1 overexpression

27) L270-271: B-Raf-induced NRF2 activation, thereby up-regulating NQO1 expre
ssion to accelerate tumor growth  B-Raf-associated NRF2 activation, thereb
y up-regulating NQO1 expression to stimulate tumor growth

28) 277-278: However, in tumor microenvironment, the presence of SFN blocks
T-cell-mediated immune response which is crucial for host defense mechanis
m  However, in the tumor microenvironment, SFN blocks T-cell-mediated i
mmune response crucial for immune surveillance of tumors

29) L282-284: This suggests that activation alone is not sufficient to induce
hepatoma formation.  This suggests that without prior DNA damage and
consequent oncogene/tumor suppressor gene mutation, NRF2 activation alone is
not sufficient to induce hepatoma formation.

30) L284-286 (also L341): In this context, SFN may function as a tumor promoter,
rather than as an initiator causing gene mutation,
 . In this context, SFN
functions in the promotion and/or progression stage(s), rather than acting as an
initiator causing gene mutation,

31) L327: 12,000 × g


12,000 × g
32) Legend to Figure 1: (B) ... the histologic difference between liver morphology in
DEN-treated and vehicle-treated WT mice. ... (D) using reverse transcriptase
enzyme following --> the histologic difference in liver morphology between DEN-
treated and vehicle-treated mice... (D) using reverse transcriptase following

33) Legend to Figure 3: ... (B) Corresponding quantifications of protein levels were
measured and
... (B) The quantification of protein levels was conducted using
Image J software, and

34) Legend to Figure 5: Corresponding quantification of protein levels of NRF2 was
measured,
 The quantification of protein levels of NRF2 was conducted,

35) Legend to Figure 6: Immunofluorescence staining of NQO1 in the livers of D
EN-treated mice with and without SFN administration 
Immunofluorescence
staining of NQO1 in the livers of control mice, DEN-treated mice with and wi
thout SFN administration, and mice treated with SFN alone